# Learning Quadruped Locomotion Using Differentiable Simulation

**Yunlong Song**
University of Zurich, Switzerland
song@ifi.uzh.ch

**Sangbae Kim**
MIT, USA
sangbae@mit.edu

**Davide Scaramuzza**
University of Zurich, Switzerland
sdavide@ifi.uzh.ch

**Abstract:** This work explores the potential of using differentiable simulation for learning quadruped locomotion. Differentiable simulation promises fast convergence and stable training by computing low-variance first-order gradients using robot dynamics. However, its usage for legged robots is still limited to simulation. The main challenge lies in the complex optimization landscape of robotic tasks due to discontinuous dynamics. This work proposes a new differentiable simulation framework to overcome these challenges. Our approach combines a high-fidelity, non-differentiable simulator for forward dynamics with a simplified surrogate model for gradient back-propagation. This approach maintains simulation accuracy by aligning the robot states from the surrogate model with those of the precise, non-differentiable simulator. Our framework enables learning quadruped walking in simulation in minutes without parallelization. When augmented with GPU parallelization, our approach allows the quadruped robot to master diverse locomotion skills on challenging terrains in minutes. We demonstrate that differentiable simulation outperforms a reinforcement learning algorithm (PPO) by achieving significantly better sample efficiency while maintaining its effectiveness in handling large-scale environments. Our method represents one of the first successful applications of differentiable simulation to real-world quadruped locomotion, offering a compelling alternative to traditional RL methods.
**Video:** https://youtu.be/weNq_w715xM

**Keywords:** Differentiable Simulation, Legged Locomotion

## 1  Introduction

Traditional model-free reinforcement learning (RL) often requires extensive parallelization to achieve a stable walking policy for legged robots [1]. How can legged robots master walking quickly with a few trials and errors? Recent progress in legged robot control has been largely driven by combining model-free RL with massively parallelized simulation [1, 2, 3, 4, 5, 6, 7]. Central to these advancements is the policy gradient algorithm [8, 9], where the gradient of a control policy can be calculated via $\nabla_\theta J(\theta) = \mathbb{E}_{\tau \sim \pi_\theta}[R(\tau)\nabla_\theta \log p(\tau; \theta)]$; which is a *zeroth-order* estimate of the true gradient of $J$ based on sampled trajectories following the policy $\pi_\theta$. Here, $R(\tau)$ is the reward of a trajectory $\tau$. Such zeroth-order gradient estimate provides a powerful and general

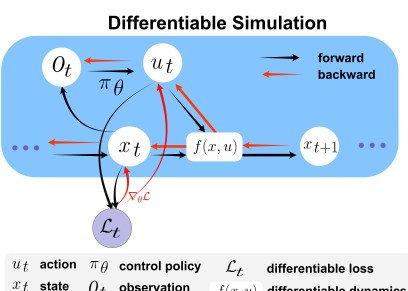

Figure 1: Graphical model for policy learning using differentiable simulation.

framework for robot control since it can handle non-differentiable optimization objectives and discontinuous dynamics. However, the gradient has high variances. Consequently, several additional strategies are required for stable training, such as using a clipped surrogate objective [10] and increasing the sample size.

8th Conference on Robot Learning (CoRL 2024), Munich, Germany.

In robotics, leveraging well-established knowledges about robot dynamics can enable the construction of *first-order* gradient estimates $\nabla_\theta J(\theta) = \nabla_\theta R(\tau)$, which typically exhibit significantly lower variance than their zeroth-order counterparts and hold great potential for more stable training and faster convergence. Recently, policy training using first-order gradient has been notably advanced through *differentiable simulation [11, 12, 13, 14, 15]*; these works have shown promising results in reducing both the number of simulation samples and total training time compared to zeroth-order methods. Differentiable simulation combines the advantages of model-based control and data-driven learning. Similar to numerical optimal control methods, such as shooting techniques, differentiable simulation utilizes robot dynamics to simulate trajectories and iteratively optimize decision variables using analytical gradients derived from the robot model. Like reinforcement learning, differentiable simulation benefits from large-scale simulations and deep learning to train neural network control policies that map observations directly to control actions. Despite its advantages, differentiable simulation faces several challenges, including discontinuous dynamics, complex optimization landscapes, and issues like exploding or vanishing gradients in long-horizon tasks, which can limit the effectiveness of first-order gradient methods.

This work investigates the potential of training policies through *surrogate models* within differentiable simulation, specifically focusing on quadruped locomotion. We show that differentiable simulation offers considerable advantages over model-free RL for policy training in legged locomotion. Notably, we demonstrate that a single robot, without the need for parallelization can quickly learn to walk within minutes in simulation using our approach. Leveraging the advantage of GPUs for parallelized simulation, our robot learns diverse walking skills over challenging terrain in minutes. Specifically, we train a quadruped robot to walk with different gait patterns, including trot, pace, bound, and gallop, and with varying gait frequencies. While model-free RL can also achieve similar performance given sufficient parallelization, our approach requires much less data, achieving significantly better sample efficiency while maintaining its effectiveness in handling large-scale environments. More importantly, we show that the policy trained via differentiable simulation can be transferred to the real world directly without fine-tuning. This work presents one of the earliest demonstrations of using differentiable simulation to train control policies for real quadruped robots, highlighting the potential of differentiable simulation for real-world applications.

**Contribution:** The key to our approach is a novel policy training framework that combines the smooth gradients obtained from a simple surrogate dynamics model for efficient backpropagation with the high fidelity of a more complex, non-differentiable simulator for accurate forward simulation. Instead of simulating the quadruped robot using whole-body dynamics, which by definition has discontinuities due to contact, we propose separating the simulation into its floating base space and its joint space. For the robot body, we employ an approximation using single rigid-body dynamics, which offers a continuous and effective representation of the robot body. To address simulation errors arising from our simplified rigid-body dynamics model, we incorporate a more precise, non-differentiable simulator. This non-differentiable simulator can simulate complex contact dynamics and is used to align the state in our simplified model, thereby ensuring that our training pipeline remains grounded in realistic dynamics. Figure 2 provides an overview of our system.

## 2 Methodology

### 2.1 Problem Formulation

We formulate legged robot control as an optimization problem. The robot is modeled as a discrete-time dynamical system, characterized by continuous state and control input spaces, denoted as $\mathcal{X}$ and $\mathcal{U}$, respectively. At each time step $k$, the system state is $x_k \in \mathcal{X}$, and the corresponding control input is $u_k \in \mathcal{U}$. An observation $o_k \in \mathcal{O}$ is generated at each time step based on the current state $x_k$ through a sensor model $h : \mathcal{X} \to \mathcal{O}$, such that $o_k = h(x_k)$. The system's dynamics are governed by the function $f : \mathcal{X} \times \mathcal{U} \to \mathcal{X}$, which describes the time-discretized evolution of the system as $x_{k+1} = f(x_k, u_k)$. At each time step $k$, the robot receives a cost signal $l_k = l(x_k, u_k)$, which is a function of the current state $x_k$ and the control input $u_k$. The control policy is represented as a

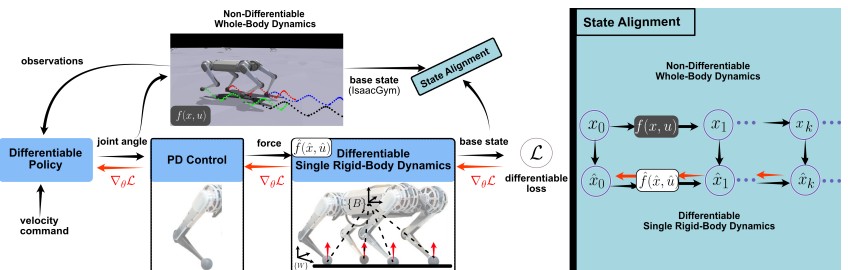

Figure 2: **System overview of learning quadruped locomotion using differentiable simulation.** Our approach decouples the robot dynamics into two separate spaces: joint and floating base spaces. We leverage the differentiability and smoothness of a single rigid-body dynamics for the robot's main body, which takes the ground reaction force from its legs as the control inputs. We use the state from a non-differentiable simulator (IsaacGym) to align the state in the differentiable simulation.

deterministic, differentiable function such as a neural network $u_k = \pi_\theta(o_k)$. The neural network takes the observation $o_k$ as input and outputs the control input $u_k$. The optimization objective is to find the optimal policy parameters $\theta^*$ by minimizing the total loss via gradient descent

$$\min_\theta \mathcal{L}_\theta = \sum_{k=0}^{N-1} l(x_k, u_k) = \sum_{k=0}^{N-1} l(x_k, \pi_\theta(o_k)) \tag{1}$$

$$\theta \leftarrow \theta - \alpha \nabla_\theta \mathcal{L}_\theta, \tag{2}$$

where $\alpha$ is the learning rate and $l(x_k, u_k)$ is the differentiable loss at simulation time step $k$.

## 2.2 Forward Simulation

We represent the robot's main body using single rigid-body dynamics. Single rigid-body dynamics have been shown to be useful for dynamic locomotion using model predictive control [16, 17]. The single rigid-body model offers a continuous representation of the robot base dynamics, avoiding the complex optimization landscape introduced by contacts. We develop our differentiable simulation using the single rigid-body dynamics, which is expressed as follows,

$$\dot{\mathbf{p}}_{WB} = \mathbf{v}_{WB} \qquad\qquad \dot{\mathbf{v}}_{WB} = \frac{1}{m}\sum_i \mathbf{f}_i + \mathbf{g}$$

$$\dot{\mathbf{q}}_{WB} = \frac{1}{2}\Lambda(\boldsymbol{\omega}_B)\cdot\mathbf{q}_{WB} \qquad\qquad \dot{\boldsymbol{\omega}}_B = \mathbf{I}^{-1}\left(\boldsymbol{\eta} - \boldsymbol{\omega}_B \times (\mathbf{I}\boldsymbol{\omega}_B)\right).$$

In this approximation, the state of our system is $\mathbf{x} = [\mathbf{p}, \mathbf{q}, \mathbf{v}, \boldsymbol{\omega}]$, where $\mathbf{p}_{WB} \in \mathbb{R}^3$ is the position and $\mathbf{v}_{WB} \in \mathbb{R}^3$ is the linear velocity of the center of mass in the world frame $W$, We use a unit quaternion $\mathbf{q}_{WB}$ to represent the orientation of the body in the world frame and use $\boldsymbol{\omega}_B$ to denote the body rates in the body frame $B$. Here, $\mathbf{I}$ is the robot's inertia tensor, $\boldsymbol{\eta}$ is the body torque, and $\Lambda(\boldsymbol{\omega}_B)$ is a skew-symmetric matrix. The control inputs are the ground reaction force $\mathbf{f}_i$ from the legs that have contacts.

The ground reaction forces are required to simulate the rigid-body dynamics. One option for the control policy design is to output the ground reaction force directly, similar to the MPC design [16, 17]. Another option is to control the robot in the joint space, e.g., output desired joint position, which allows more control authority for the policy and adaptive behavior. Our policy maps observations to the desired joint position $\mathbf{q}^{\text{ref}}$, while assuming zero joint velocity, i.e., $\dot{\mathbf{q}}^{\text{ref}} = \mathbf{0}$. In this case, the neural network output (joint position) is required to be converted into the control input of the single rigid-body model, which is the ground reaction force. This conversion is achieved using a PD controller for forward propagation

$$\boldsymbol{\tau} = \mathbf{k_p}(\mathbf{q}^{\text{ref}} - \mathbf{q}) + \mathbf{k_d}(\dot{\mathbf{q}}^{\text{ref}} - \dot{\mathbf{q}}), \tag{3}$$

which calculates the required motor torque $\boldsymbol{\tau}$. Here, $\mathbf{k_p}$ and $\mathbf{k_d}$ are fixed gains, $\mathbf{q}^{\text{ref}}$ and $\mathbf{q}$ are the reference joint position and the current joint position separately, and $\dot{\mathbf{q}}^{\text{ref}}$ and $\dot{\mathbf{q}}$ are the reference

joint velocity and the current joint velocity respectively. Subsequently, the motor torques are then converted to ground reaction forces $\mathbf{f}$ using the foot Jacobian $\mathbf{J}$: $\mathbf{f} = (\mathbf{J}^T)^{-1}\boldsymbol{\tau}$. The continuous nature of PD control enables the backpropagation of policy gradients. Consequently, our simulation framework treats the PD controller as a differentiable layer.

## 2.3 Backpropagation Through Time

In differentiable simulation for policy learning, the backward pass is crucial for computing the analytic gradient of the objective function with respect to the policy parameters. Following [18], the policy gradient can be expressed as follows

$$\nabla_\theta \mathcal{L}_\theta = \frac{1}{N} \sum_{k=0}^{N-1} \left( \sum_{i=1}^{k} \frac{\partial l_k}{\partial x_k} \underbrace{\prod_{j=i}^{k} \left( \frac{\partial x_j}{\partial x_{j-1}} \right)}_{\text{differentiable dynamics}} \frac{\partial x_i}{\partial \theta} + \frac{\partial l_k}{\partial u_k} \frac{\partial u_k}{\partial \theta} \right), \tag{4}$$

where the matrix of partial derivatives $\partial x_j / \partial x_{j-1}$ is the Jacobian of the dynamical system $f$. Therefore, we can compute the policy gradient directly by backpropagating through the differentiable physics model and a loss function $l_k$ that is differentiable with respect to the system state and control inputs. A graphical model for gradient backpropagation in policy learning using differentiable simulation is given in Figure 1. Due to the usage of multiplication $\prod$, there are two potential issues in using Eq. (4) for policy gradient: 1) gradient vanishing or exploding, 2) long computation time. We tackle these two problems via short-horizon policy training.

## 2.4 State Alignment with A Non-Differentiable Simulator

Due to the simplification of our single rigid-body dynamics and the decoupling of the simulation space, the robot state can diverge from its actual one. Over time, even minor discrepancies can accumulate, leading to unrealistic states and ultimately causing the failure of policy training. We proposed to align the body state in our differentiable simulation using information from other simulators that use accurate whole-body dynamics. Specifically, we align the robot state in our differentiable simulation with the state information from IsaacGym [19].

We use the following equation to align the robot state $\hat{\mathbf{x}}_{t+1}^{\text{diff}} = \mathbf{x}_{t+1}^{\text{non-diff}} + \alpha * (\mathbf{x}_{t+1}^{\text{diff}} - \mathbf{x}_{t+1}^{\text{diff, detach}})$. Here, $\mathbf{x}_{t+1}^{\text{diff, detach}}$ and $\mathbf{x}_{t+1}^{\text{diff}}$ represent the same robot state of our differentiable simulator at time step $t + 1$; hence, they share the same value. The word *detach* indicates that $\hat{\mathbf{x}}_{t+1}^{\text{diff, detach}}$ is detached from the computational graph for automatic differentiation. Therefore, we can reset the robot state using the value from the non-differentiable simulation during forward simulation: $\hat{\mathbf{x}}_{t+1}^{\text{diff}} = \mathbf{x}_{t+1}^{\text{non-diff}} + \alpha * \mathbf{0}$. During backpropagation, the gradient of the state at a given time $t$ is computed as follows $\partial \hat{\mathbf{x}}_{t+1}^{\text{diff}} / \partial \mathbf{x}_t^{\text{diff}} = \mathbf{0} + \alpha * \partial \mathbf{x}_{t+1}^{\text{diff}} / \partial \mathbf{x}_t^{\text{diff}}$. Here, $0 < \alpha \le 1$ is used to decay the gradient.

The non-differentiable simulator can simulate complex contact dynamics and is used to align our simplified model, thereby ensuring that our differentiable training pipeline remains grounded in realistic dynamics. At the same time, our approach benefits from the simplified differentiable simulator, which offers smooth gradients for backpropagation. Figure 2 shows the computational graph of the forward propagation and backpropagation using state alignment.

## 2.5 Short-Horizon Policy Training

Although smoothed physical models (e.g., single rigid-body dynamics) improve the local optimization landscape, the complexity of the optimization problem escalates significantly in long-horizon problems involving extensive concatenation of simulation steps. The situation further deteriorates when the actions within each step are interconnected through a nonlinear and nonconvex neural network control policy. The complexity of the resulting optimization landscape can make gradient-based methods to become trapped in local optima quickly. Instead of directly solving a long-horizon policy training problem, following [13, 1], where long-horizon simulation tasks are truncated into

short-horizon simulation, we utilize short-horizon policy learning to achieve stable gradient back-propagation over a short horizon. Mathematically, short-horizon policy learning involves truncating a long-horizon trajectory of length $N$ into several shorter segments to compute the policy gradient shown in Eq. (4). For instance, for a trajectory of length 240, we divide it into ten segments of $N = 24$. However, it's important to note that while a shorter horizon could simplify computation and facilitate optimization, it may also limit the model's ability to anticipate future events, introducing bias into the gradient calculation. This bias could, in turn, affect the types of tasks the model can effectively learn.

## 2.6 Differentiable Loss Function

A fundamental difference between policy training using differentiable simulation and reinforcement learning is that the loss function has to be differentiable when using differentiable simulation. RL allows the direct optimization of non-differentiable rewards, such as binary rewards $0/1$. On the contrary, differentiable simulation requires a smooth differentiable cost function to provide learning signals for the desired control inputs. We formulate a differentiable loss function $l(\mathbf{x}_t, \mathbf{u}_t)$ tailored for velocity tracking, where the main objective is to follow a specified velocity command, denoted as $\mathbf{v}^{\text{ref}}$. Additionally, we maintain the robot's body height, represented by $p_z$. To enhance the robustness of this system, we incorporate several regularization terms: one to mitigate a large angular velocity, thus controlling the body rate $\boldsymbol{\omega}$; another to limit the output action $\mathbf{u}$, preventing large control actions; and a term aimed at stabilizing the robot's orientation using projected gravity vector $\mathbf{g}_{\text{proj}}$. The loss function is defined as

$$
\begin{aligned}
l(\mathbf{x}_t, \mathbf{u}_t) \;=\; & a_1\|\mathbf{v} - \mathbf{v}^{\text{ref}}\|^2 + a_2\|p_z - p_z^{\text{ref}}\| + a_3\|\boldsymbol{\omega}\|^2 \\
& + a_4\|\mathbf{u}\|^2 + a_5\|\mathbf{g}_{\text{proj}}\| + a_6\|\mathbf{p}_{\text{foot}} - \mathbf{p}_{\text{foot}}^{\text{ref}}\|^2,
\end{aligned}
\tag{5}
$$

where $\mathbf{p}_{\text{foot}}$ is the foot position.

## 3 Experimental Results

### 3.1 A Toy Example: Control of A Double Integrator

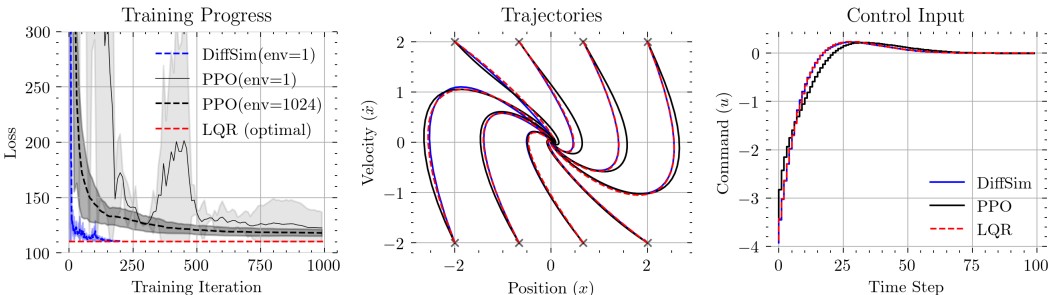

Figure 3: **Control of a double integrator using optimal control, reinforcement learning, and differentiable simulation.** (left): Learning curves. (middle): Trajectories of different control policies. We initialize the system at the same states for all methods. (right): DiffSim achieves control commands close to optimal control.

Inspired by [20], where the author studied the connection between reinforcement learning and optimal control using a toy double integrator example, we begin by examining the same toy example: control of a double integrator, which is a fundamental problem in system and control theory. The double integrator is a second-order control system that models simple point mass dynamics in one-dimensional space. The state variables are position $x$ and velocity $\dot{x}$, with control inputs $u = \ddot{x}$. The objective is to drive the system state from any initial point to the origin, $[x, \dot{x}] = [0, 0]$. Therefore, this problem can be solved by the Linear quadratic regulator (LQR), which can provide an optimal solution. We train a 2-layer multiplayer perception using backpropagation through time via differentiable simulation and the proximal policy optimization (PPO) method [10]. We show that the

policy trained using differentiable simulation (DiffSim) achieves nearly optimal performance within a handful of training iterations and samples. In contrast, even when scaling the number of simulation environments to 1024 and training the policy with considerably more iterations, PPO fails to achieve the same level of control performance as LQR or DiffSim. This toy example suggests that scaling might be insufficient for PPO to achieve an optimal solution.

## 3.2   Learning to Walk with One Robot

This section explores whether a robot can learn to walk without parallelization and with very limited data points at each training iteration. We design a simple velocity-tracking task where the robot is required to follow a constant velocity in the $x$-axis, e.g., $v_x = 0.2 \ m/s$. At each training iteration, we simulate 24-time steps with one single robot. Hence, each training iteration contains 24 data points. We trained the policy for a total of 1000 iterations. The result is given by the learning curve in Figure 4. Despite very limited data, our policy successfully learns to walk after minutes of training. As a comparison, PPO failed to achieve useful locomotion skills under the same condition. By providing first-order gradients with low variance, differentiable simulations offer a fundamental advantage: they allow for more

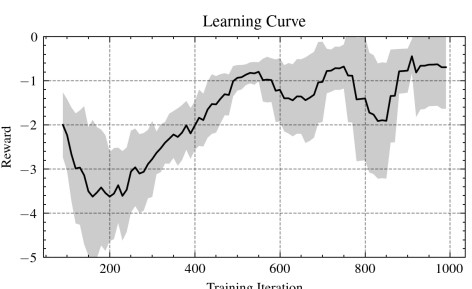

Figure 4: **Learning to walk with one simulated robot.** We run 10 experiments with different random seeds. The plot is smoothed using a moving average.

efficient and reliable optimization of the control policy. This is because the gradient provides a clear direction for updating parameters, and the low variance ensures that this direction is consistent and reliable, even when data is limited.

## 3.3   Learning Diverse Walking Skills on Challenging Terrains

We design a more difficult task: learning diverse locomotion skills over challenging terrains. Specifically, we design four different gait patterns, including *trot*, *pace*, *bound*, and *gallop*. Additionally, we vary the gait frequency from $1 \ \mathrm{Hz}$ to $4 \ \mathrm{Hz}$. The robot receives a high-level velocity command and is required to track randomly commanded velocity. Figure 5 shows the learning curves using different numbers of robots for policy training. We compare the learning performance of our method with a model-free RL algorithm (PPO) [10]. Given minimal samples, e.g., only four robots, RL has a slow convergence speed and does not learn meaningful walking skills, e.g., the policy constantly falls after a few simulation steps. In contrast, differentiable simulation achieves much higher rewards and can acquire useful walking skills.

As the number of robots increases, the performance of both algorithms improves. Notably, the performance improvement for RL is much more significant than our approach. This indicates that the zeroth-order gradient estimates used by reinforcement learning are generally inaccurate and require many more samples to achieve stable training. On the contrary, the first-order gradient estimates used by our differentiable simulation can have very stable and accurate gradients, even given very limited simulation samples. Figure 5 demonstrates diverse walking skills over challenging terrains using a blind policy trained via differentiable simulation.

We demonstrate the performance of our policy in the real world using Mini Cheetah [21]. Mini Cheetah is a small and inexpensive, yet powerful and mechanically robust quadruped robot, intended to enable the rapid development of control systems for legged robots. The robot uses custom back-driveable modular actuators, which enable high-bandwidth force control, high force density, and robustness to impacts [21]. The control policy runs at $100 \ \mathrm{Hz}$ during deployment. Figure 5 shows several snapshots of the robot's walking behavior over different terrains using different gait patterns. We trained a blind policy in simulation using 64 robots and then transferred the policy directly to

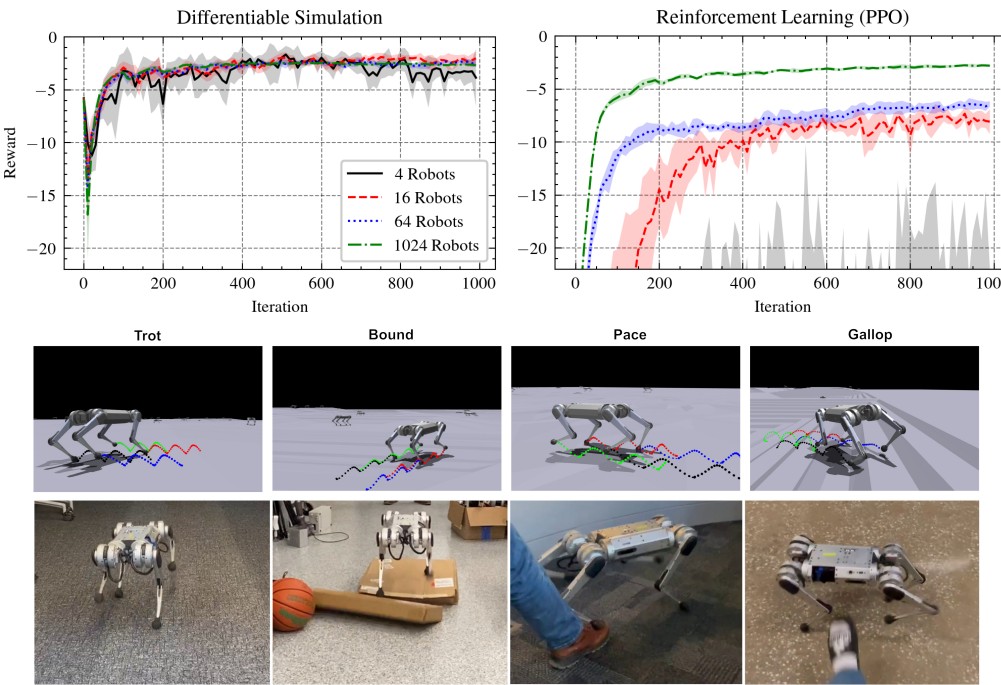

Figure 5: **Learning to walk on challenging terrains, reinforcement learning versus differentiable simulation.** Differentiable simulation exhibits significant advantages over PPO in terms of sample efficiency and learning stability. After training in simulation, the policy can be transferred to the real world without fine-tuning.

the real world without fine-tuning. The robot can walk forward and backward with different gait patterns and frequencies. Moreover, the policy enables the robot to manage disturbances, such as unexpected forces and deformable terrain.

Additionally, we compare the total training wall-clock time and the final performance of the resulting policies. Differentiable simulation often involves the creation of a complex computational graph to facilitate gradient computation. As the simulation horizon extends, the length of the computational graph can increase proportionally, leading to a substantial increase in the total training time. This increase can diminish its bene-

| Robots | Final Reward | | Total Training Time [min] | |
|---|---|---|---|---|
| | PPO | **Ours** | PPO | **Ours** |
| 4 | $-25.82 \pm 2.16$ | $\mathbf{-3.83 \pm 3.01}$ | $7.27 \pm 0.12$ | $7.28 \pm 0.12$ |
| 16 | $-8.07 \pm 1.30$ | $\mathbf{-2.03 \pm 0.69}$ | $7.54 \pm 0.06$ | $7.50 \pm 0.08$ |
| 64 | $-6.42 \pm 0.29$ | $\mathbf{-2.49 \pm 0.37}$ | $8.46 \pm 0.16$ | $8.23 \pm 0.07$ |
| 1024 | $-2.80 \pm 0.13$ | $\mathbf{-2.69 \pm 0.06}$ | $11.44 \pm 0.13$ | $11.55 \pm 0.09$ |

Table 1: **A comparison of final reward and total training time.** We run both methods for the same number of training iterations and collect the same amount of samples.

fits in situations where gathering a large number of samples is both cheap and fast. Therefore, we ask the question: *Can differentiable simulation be effectively applied to large-scale settings, for example, thousands of environments?* To ensure a fair comparison, we ran both PPO and our training framework for the same number of training iterations, namely 1000 iterations. We compare the total training time, which includes the time for simulating dynamics and updating policies. As shown in Table 1, our results indicate that differentiable simulation can be as time efficient as PPO in large-scale environments despite the requirement of backpropagation through time (BPTT)—a process known for its computational demands. However, notice that the training time to reach a certain reward is much lower with our method and requires less robot and data. For example, our method requires only roughly 64 robots and 200 training iterations to solve this task, while PPO still performs less after 1000 training iterations with 1024 robots.

# 4 Related Work

Previous efforts in leveraging differentiable simulation to robot control can be broadly categorized into two classes: improving the simulator itself and enhancing the associated optimization algorithms. The first kind has led to the development of several general-purpose differentiable simulators, including Brax [11], Nimble [22], NeuralSim [23], and Dojo [24]. These simulators enable the integration of robot models directly into policy optimization, allowing neural network control policies to be trained using more mathematically precise analytical gradients. However, as noted in [18, 25], in scenarios where the underlying system exhibits chaotic behavior—where small perturbations in initial conditions lead to significantly divergent states, such as in robotic simulations involving contact dynamics—these gradients can become unmanageable, leading to divergence in optimization processes. Our work highlights the benefits of employing well-behaved proxy dynamics as an alternative to the true robot dynamics involving contact. Such a design choice has its roots in model predictive control, which often leverages a simplified model for the optimization to control complex robotic systems [16, 17, 26].

Besides developing a better simulator, various methods have been proposed to improve policy training algorithms. For example, [25] proposes an $\alpha$-order gradient estimator that aims to combine the efficiency of first-order estimates with the robustness of zeroth-order methods. SHAC [13] addresses local minima issues by learning a smooth value function and mitigates vanishing/exploding gradients through truncated backpropagation. Gradient norm clipping strategies have also proven effective in managing the exploding gradients issue [27]. Furthermore, since differentiating through chaotic dynamics is challenging, it is often useful to learn an approximation of these functions and use the approximation for gradient computation. This approach commonly has a connection with model-based reinforcement learning [28, 29, 30, 31, 32], which involves learning a predictive model of the environment, followed by training a controller. Similar to [13], our method leverages a truncated backpropagation to facilitate optimization. Our method could be improved further by combining other ideas, such as learning a critic function or clipping the gradients.

# 5 Limitations

A key advantage of model-free RL lies in its ability to directly optimize task-level rewards [33], especially non-differentiable rewards. In contrast, differential simulation depends on the backpropagation of information from its objective function, making it challenging to explore novel solutions guided by task objectives. Consequently, our system requires specifying foot positions using the Raibert heuristic [34], as it cannot explore foot placement motion through velocity tracking loss alone. In addition, RL's flexibility allows for more robust performance enhancements, such as implementing simple termination penalties to discourage falling behaviors or employing constant survival rewards to encourage continued operation. Furthermore, differentiable simulations are often tailored to specific optimization tasks and might not generalize well across different problems. It requires specific domain knowledge in both building the simulation pipeline and formulating the problem. To generalize our approach to other tasks, additional engineering efforts are required.

# 6 Conclusion

This work proposes a new perspective on learning quadruped locomotion policies using differentiable simulation by integrating a simplified rigid-body model with a non-differentiable model for whole-body dynamics. Our key insight is that a neural network control policy can be optimized directly by backpropagating the differentiable loss function through a simplified model while simultaneously aligning the robot's state using a more accurate whole-body dynamics model. Future work should address several limitations of the proposed framework, such as removing the assumption of gait scheduling and the Raibert heuristic for foot contact sequence and foot position planning. Additionally, more sim-to-real analysis is helpful in evaluating the benefits of using a simplified model compared to the true but chaotic gradients or to the policy trained via domain randomization. Finally, we should demonstrate the true advantages of stable training introduced by the first-order gradient by tackling more challenging tasks, such as large-scale training and vision-based control.

# 7 Acknowledgment

This work was supported by the European Union's Horizon Europe Research and Innovation Programme under grant agreement No. 101120732 (AUTOASSESS) and the European Research Council (ERC) under grant agreement No. 864042 (AGILEFLIGHT). The authors especially thank Yuang Zhang for his help with the differentiable simulation development and for the useful discussion, Seungwoo Hong for the useful discussion about rigid-body dynamics and nonlinear model predictive control, and Nico Messikommer for his valuable feedback. Also, the authors thank Se Hwan Jeon, Ho Jae Lee, Matthew Chignoli, Steve Heim, Elijah Stanger-Jones, and Charles Khazoom for their helpful support.

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

## Appendix

## 1 Double Integrator

A double integrator system is characterized by its position and velocity. The control input $u$ directly controls the acceleration of the system. For a fair comparison, we formulate the problem as a discrete-time finite-horizon optimal control problem. The discrete-time state space representation of a double integrator is

$$x_{k+1} = Ax_k + Bu_k$$

where $A = \begin{bmatrix} 1 & d_t \\ 0 & 1 \end{bmatrix}$ and $B = \begin{bmatrix} d_t^2/2 \\ dt \end{bmatrix}$. Here, $d_t$ is the simulation time step. The objective is to minimize a quadratic loss function

$$J = x_N^T Q_f x_N + \sum_{k=0}^{N-1} (x_k^T Q x_k + u_k^T R u_k)$$

$$Q = Q_f = \begin{bmatrix} 1 & 0 \\ 0 & 1 \end{bmatrix}, R = [1].$$

This is a discrete-time finite-horizon Linear–Quadratic Regulator problem, where the optimal control law can be found using dynamic programming

$$u_k^* = -(R + B^T P_{k+1} B)^{-1} B^T P_{k+1} A x_k, \quad k = 0, 1, \cdots, N-1,$$

where $P_N = Q_f$ and $P_k$ can be found from the Riccati recursion.

## 2 Experimental Setup

**Simulation Setup:** We develop our own differentiable simulator using PyTorch and CUDA. Our differentiable simulator allows for both forward propagation of the robot dynamics and backpropagation of the policy gradient. Additionally, we run IsaacGym [19] alongside our differentiable simulation and use it to align the robot state resulting from our simplified robot dynamics. Isaac-Gym simulates the whole-body dynamics and complicated contacts between the robot and its environment. Both simulations are parallelized on GPU. We used a discretized simulation time step of $0.002\,\text{s}$ and a control frequency of $100\,\text{Hz}$. We use PPO baseline from [1]. Additionally, we follow prior works [35, 36] for the implementation of the gait schedule and Raibert Heuristic.

**Observation and Action:** The policy observation includes random commands ($\mathbf{cmd}_{\text{rand}}$) for the reference velocity, sinusoidal and cosinusoidal representations of gait phases, the base velocity ($\mathbf{v}_{WB}$), the base orientation ($\mathbf{q}_{WB}$), the angular velocity ($\boldsymbol{\omega}_B$), motor position deviations from default ($\mathbf{q} - \mathbf{q}_{\text{default}}$), and a projected gravity vector ($\mathbf{g}_{\text{projected}}$). The policy action $\delta\mathbf{q}$ is the desired joint position offset from the default joint position.

| Observation | Dimension | Action | Dimension |
|:---:|:---:|:---:|:---:|
| $\mathbf{cmd}_{\text{rand}}$ | 3 | | |
| $\mathbf{sin}$(gait phase) | 4 | | |
| $\mathbf{cos}$(gait phase) | 4 | | |
| $\mathbf{v}_{WB}$ | 3 | $\delta\mathbf{q}$ | 12 |
| $\mathbf{q}_{WB}$ | 4 | | |
| $\boldsymbol{\omega}_B$ | 3 | | |
| $\mathbf{q} - \mathbf{q}_{\text{default}}$ | 12 | | |
| $\mathbf{g}_{\text{projected}}$ | 3 | | |

Table 2: Policy observation and action for quadruped locomotion.

## 3 Foot Trajectory Planning

We use the foot position loss $\|\mathbf{p}_{\text{foot}} - \mathbf{p}_{\text{foot}}^{\text{ref}}\|^2$ to provide a learning signal for the swing legs. This loss term is critical for the swing leg since it contains information about the motor position. Following prior works [35, 36], the swing leg trajectory can be computed by fitting a quadratic polynomial over the lift-off $\mathbf{p}_{\text{foot}}^{\text{lift}}$, mid-air $\mathbf{p}_{\text{foot}}^{\text{air}}$, and landing position $\mathbf{p}_{\text{foot}}^{\text{land}}$ of each foot, where the lift-off position is the foot location at the beginning of the swing phase, the landing position $\mathbf{p}_{\text{foot}}^{\text{land}}$ is calculated using the Raibert Heuristics [34], which is expressed as the following function

$$\mathbf{p}_{\text{foot}}^{\text{land}} = \mathbf{p}_{\text{foot}}^{\text{hip}} + \mathbf{v}^{\text{CoM}} T_{\text{stance}}/2$$

where $T_{\text{stance}}$ is the expected time the foot will spend on the ground, $\mathbf{p}_{\text{foot}}^{\text{hip}}$ is the location on the ground beneath the robot's hip, $\mathbf{v}^{\text{CoM}}$ is the body velocity projected on the $xy$-plane. The desired contact state of each leg. e.g., swing or stance, is determined via a gait generator. The gait is modulated by a phase variable $\phi \in [0, 2\pi]$. The phase is defined through a dynamic function $\phi_{t+1} = \phi_t + 2\pi f \Delta t$, where $f$ is the stepping frequency. We can design different locomotion patterns by adapting the stepping frequency $f$ and the phase difference between each leg.

## 4 On the Importance of Non-differentiable Terminal Penalty

We highlight one important benefit of RL compared to differentiable simulation: RL can significantly enhance its robustness by directly optimizing through non-differentiable rewards or penalties. Specifically, we use a non-differentiable value $p = 200$ to penalize the robot when the robot experiences termination during training, e.g., falling on the ground or lifting its legs above its body.

$$r(\mathbf{x}_t, \mathbf{u}_t) = \begin{cases} -l(\mathbf{x}_t, \mathbf{u}_t) - p & \text{if termination} \\ -l(\mathbf{x}_t, \mathbf{u}_t) & \text{otherwise.} \end{cases}$$

Fig. 6 shows a study of using non-differentiable terminal penalty for both RL and differentiable simulation. The results show that adding a final penalty can greatly affect how well RL works. Without a penalty, RL might get trapped in a local minimum. However, with a large penalty at the end, RL can achieve better task rewards as well as more robust control performance. This is because RL optimizes a discounted return, which estimates "how good" it is to be in a given state. RL uses a state-value function to encode this information

$$V_\pi(s) = \mathbb{E}[G|S_0 = s] = \mathbb{E}\left[\sum_{t=0}^{\infty} \gamma^t R_{t+1}|S_0 = s\right].$$

On the other hand, a terminal penalty has no impact on differentiable simulation since the gradient of a constant value is equal to zero, and we do not leverage a state value function. As a result, differentiable simulation requires well-defined continuous functions, e.g., a potential function or control barrier functions for robust control.

## 5 Ablation Study

We conducted an ablation study to investigate the significance of the proposed state alignment mechanism. In our implementation, we use the state from IsaacGym to align the robot state within our simplified rigid-body differentiable simulation. To assess the impact of state alignment, we performed experiments where we removed the state alignment and compared the resulting learning curves to those obtained with state alignment. The results are presented in Figure 7. The results indicate that without state alignment, the robot fails to learn any useful walking skills.

## 6 Hyperparameters

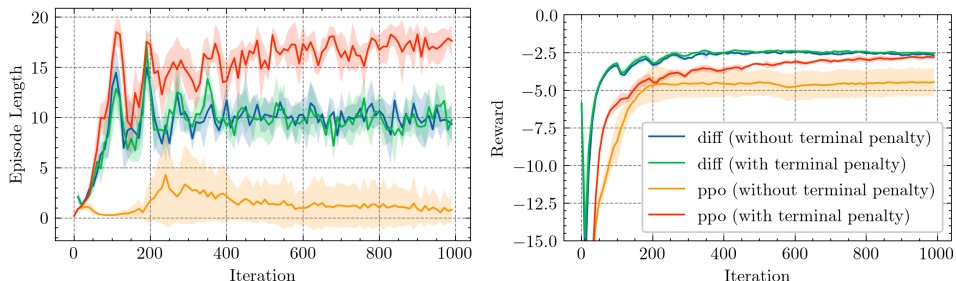

Figure 6: **A comparison of non-differentiable terminal penalty for policy training.** Using a non-differentiable terminal penalty, PPO can achieve robust control performance, e.g., longer episode length. We use 1024 robots for simulation.

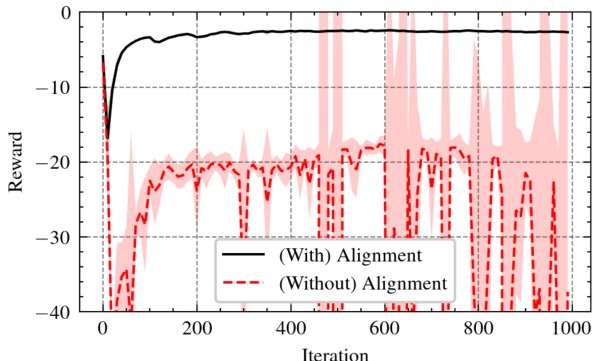

Figure 7: Ablation study for state alignment using IsaacGym.

| Parameter | Value |
|---|:---:|
| learning rate | 0.001 |
| discount factor $\gamma$ | 0.95 |
| GAE-$\lambda$ | 0.95 |
| learning epoch | 10 |
| policy network | MLP [256, 256] |
| value network | MLP [256, 256] |
| clip range | 0.2 |
| entropy coefficient | 0.002 |
| number of epoch | 10 |

Table 3: PPO hyperparameters.

| Parameter | Value |
|---|:---:|
| learning rate | 0.001 |
| policy network | MLP [256, 256] |
| gradient decay factor $\alpha$ | 0.9 |

Table 4: Hyperparameters for policy training using differentiable simulation.

