# OpenReview forum: "Learning Quadruped Locomotion Using Differentiable Simulation"
_robot-learning.org/CoRL/2024/Conference — CoRL 2024_

### Official Review · Reviewer_VLv4 · 2024-07-19

**Originality:** 4
**Technical Quality:** 4
**Clarity Of Presentation:** 4
**Potential Impact:** 4
**Recommendation:** 4
**Confidence:** 4

**Review:**

**Overview**

This paper makes a pioneering attempt in using differentiable simulation (or more broadly, physical priors) to reduce the variance in reinforcement learning. While the proposed method may not fully operate at the scale of non-differentiable RL yet, it provides valuable insight to the field.

**Strengths**

* Effective dynamics model choice

I like the author's choice of the single-rigid-body (SRBD) model as the main model for differentiable simulation. Compared to the full dynamics model, this model is much simpler to implement and compute, while still being highly accurate.

* Learning framework design

The authors carefully designed many key components of the learning framework, such as the iterative alignment between differentiable and full dynamics (Eq.5-6) and the truncated backpropagation. These carefully designed components contribute to the overall effectiveness of the method.

* Clear, insightful results section

The entire paper, and the results section particularly, is very well-organized and clear. I like that the authors start with a double integrator example and compared the result of PPO, DiffSim and LQR. The following sections further demonstrates the framework's performance and addresses important concerns such as training time.

**Weaknesses**

* The proposed simulator may not operate at the scale of "massive end-to-end RL" yet

The proposed learning framework still relies on a few key "signals" to "aid" learning, such as the reference foot position and the gait phases. These guidances might limit the expressiveness of the policy, and prevents the robot from learning more complex behaviors (e.g. using legs to manipulate objects). It would be great if the authors could study some of these in future works.

* More sim-to-real analysis would be desirable

While the authors claim that the diff-sim-trained policy can be transferred directly to the real world, it would be interesting to further analyze on the sim-to-real gap beetween different policies. Would the additional gradient signal reduce, or increase the sim-to-real gap, compared to a PPO policy trained with/without domain randomization? That would be another way to extend the impact of this paper.

**Quality Of The Limitations Section:**

3

**Questions For Rebuttal:**

I have a few detailed questions/comments:

1. Eq.(3) seems to suggest that the desired action space is both the reference joint position/velocity, while the actual action space seems to be position only (from the appendix)? Can you clarify this in the revision?

2. Are gaits / stepping frequencies the *commands* to the policy, or are those figured out by the policy itself? Would the same framework automatically determine the contact sequences/frequencies?

3. How is contact determined in the SRBD model? That is, how does the model know which legs provide GRFs and which legs don't? Is it based on the reference gait, or the simulator's contact detection? How would this contact detection affect the learning process?

4. It would be great to discuss the "short-horizon policy training" in detail. How short is the horizon used? How does this horizon affect the training time and policy performance?

5. It seems that you are using vanilla policy gradient (Eq.(4)) for diffsim, but the PPO policy gradient for non-diff-sim baseline. I understand that diffsim provides lower variance gradients, which might make the variance-reduction techniques in PPO redundant. However, would the long chains of back-propagation lead to any vanishing/exploding gradients? In these cases, would gradient-clipping techniques in PPO help in stabilizing training?

**Robotics Focus:**

4

**Summary Of Paper:**

This paper presents a method that uses a differentiable simulator to aid the gradient computation in RL. The differentiable simulator adopts a simpler dynamics model and is iteratively aligned with the full dynamics. The result shows efficient learning of high-performance policies.

**Summary Of Recommendation:**

Overall a great paper with interesting new ideas. It should be one of the top papers this year.

---

### Official Review · Reviewer_ersp · 2024-07-20
**Well-designed method and good execution**

**Originality:** 4
**Technical Quality:** 3
**Clarity Of Presentation:** 4
**Potential Impact:** 3
**Recommendation:** 3
**Confidence:** 3

**Review:**

**Strength:**

- Originality: I find the reformulation of dynamics into two differentiable components novel and well-designed. However, due to the poor quality of *Related Work* section, I am not certain of my evaluation here.
- Clarity: The paper is polished and well-written, except for the *Related Work* section.
- Quality: The paper is well-executed and the experiments are well-thought-out. I especially like the evaluation of the wall-clock time and appreciate the zero-shot real world transfer.

**Weakness:**

- The quality of writing for the *Related Work* section is disparate from the rest of the paper. The content is generic and kind of off-the-point.  It should discuss existing works on differentiable simulator. As an example, CALIPSO also demonstrated its use on quadruped gait, albeit in simulation.

	*Howell, Taylor A., Kevin Tracy, Simon Le Cleac’h, and Zachary Manchester. "CALIPSO: A differentiable solver for trajectory optimization with conic and complementarity constraints." In The International Symposium of Robotics Research, pp. 504-521. Cham: Springer Nature Switzerland, 2022.*


    It should also cover existing knowledge on the use zeroth vs. first-order policy gradients, such as:


	*Suh, Hyung Ju, Max Simchowitz, Kaiqing Zhang, and Russ Tedrake. "Do differentiable simulators give better policy gradients?." In International Conference on Machine Learning, pp. 20668-20696. PMLR, 2022.*

- Baseline: The paper only has one PPO baseline. Aside from model-free RL, the paper should compare its tailored solution to existing general-purpose differentiable simulators.

Minor comment: While $x$ and $\mathbf{x}$ refer to the same variable, $f$ and $\mathbf{f}$ refer to different variables. The letter $f$ refers to a third thing in Equation 7.

**Quality Of The Limitations Section:**

2

**Questions For Rebuttal:**

Referring to the review section:

- Improve the quality of writing of the *Related Work* section
- Choice of baselines: Compare with existing differentiable simulators. How does the differentiable simulation tailored to quadruped compare with existing general purpose differentiable simulator? How does the quality of the gradients compare?

Clarifications:

- Can you elaborate on the sources of misalignment between the simplified, differentiable simulator and the high-fidelity, non-differentiable one? My understanding is 1) linearization error from foot Jacobian, and 2) compounding error over time.
- What does blind policy (line 234) mean? Does it mean the robot does not have visual perception?
- One rollout consists of 24 time steps. Does back-propagation through time (BPTT) differentiable through all 24 time steps? Or is the computation graph detached at every k step? It would be interesting to ablate what the optimal k is.

Questions:

- In Table 1, the performance of DiffSim actually degrades beyond 16 robots. Why is that?
- I am curious about how much engineering effort is required for such a tailored solution. More importantly, how much additional effort would it take to apply the method to another quadruped, such as Spot or Go1?

**Robotics Focus:**

4

**Summary Of Paper:**

The paper proposed a novel differentiable simulation framework (DiffSim) for quadruped locomotion. To address the inherent non-differentiability of the dynamics, DiffSim decomposed it into two differentiable components. DiffSim also used a high-fidelity, non-differentiable simulator (IssacGym) for forward simulation to correct that of the simplified, differentiable simulation. DiffSim demonstrates significantly better sample efficiency compared to a model-free RL baseline, and transfers to real world zero-shot.

**Summary Of Recommendation:**

The paper is well-written, the method is interesting, and the experiments are good overall. But, it has a notable weakness of having a single baseline, and not comparing to existing differentiable simulators.

---

### Official Review · Reviewer_Mbex · 2024-07-21
**Calculate the first order policy gradient via differentiable simulation**

**Originality:** 5
**Technical Quality:** 5
**Clarity Of Presentation:** 5
**Potential Impact:** 4
**Recommendation:** 4
**Confidence:** 3

**Review:**

Zeroth order Policy Gradients have a high variance which is mitigated by various modifications shown in 'Proximal Policy Optimization'. In this paper, the authors formulate a differentiable formulation of rigid body dynamics which lets one calculate the first order policy gradient which has comparatively lower variance.

Strengths
1. Clear and concise mathematical formulation which lets non-experts (like me) to understand the paper.
2. Great experiment design of increasing complexity along with hardware trials which show clear benefits of this approach over other approaches.
3. Clear understanding of the limitations of the approach along with suggestions to mitigate those.

Weaknesses
1. The training time seems to be the same for both the approaches in Table 1. A better understanding of it would be great.
2. Having a model-based RL baseline would have been nice to show the benefits of proposed approach as well as limitations of MBRL.
3. A small ablation study to show the effect of state alignment would be interesting to show how the states drift from the real simulator.

**Quality Of The Limitations Section:**

3

**Questions For Rebuttal:**

1. It's interesting that the training time seems to be the same for both the approaches in Table 1. The authors mention "our results indicate that differentiable simulation can be as time-efficient as PPO in large-scale environments despite the requirement of backpropagation through time (BPTT)—a process known for its computational demands." It would be great if the authors could comment this is the case?

2. Was there a specific reason for choosing IsaacSim as the simulator of choice?

3. "we utilize short-horizon policy learning to achieve stable gradient back-propagation over a short horizon." a short mathematical description of it would be nice. Wouldn't the shorter horizon impact the nature of tasks learnable in this setup?

4. would the ground reaction forces also need to be aligned?

5. Having a model-based RL baseline would have been nice to show the benefits of proposed approach as well as limitations of MBRL.

6. A small ablation study to show the effect of state alignment would be interesting to show how the states drift from the real simulator.

**Robotics Focus:**

4

**Summary Of Paper:**

The authors formulate a single rigid body dynamics approximation of a legged robot which is differentiable. The reward function is differentiable with respect to the model parameters which lets us calculate the lower variance first order policy gradient. The state is aligned with a non-differentiable simulator to avoid tracking errors.

**Summary Of Recommendation:**

Well written paper well thought out experiments to show the clear benefits of differentiable simulation for legged locomotion.

---

### Author Rebuttal · Authors · 2024-08-09

Dear Reviewers,

We have provided a detailed reply to the individual comments and improved our manuscript. We highly appreciate the encouraging positive comments and the useful feedback from three reviewers. Please find our reply to each individual comment below and in the revised manuscript.

---

### Decision · Program_Chairs · 2024-09-04

**Decision:**

Accept

**Comment:**

The paper presents a differentiable simulation framework for learning legged locomotion, which tries to overcome the issues of discrete dynamics. The reviewers found that the paper is well written and conveys impressive technical insights, which may have a huge impact on the follow-up works. However, there are several clarification requests from the reviewers- please refer to the individual comments.

The paper has been unanimously supported by reviewers, and the AC also recommends acceptance. Congratulations.